# Social Inequalities in Health Determinants in Spanish Children during the COVID-19 Lockdown

**DOI:** 10.3390/ijerph18084087

**Published:** 2021-04-13

**Authors:** Yolanda González-Rábago, Andrea Cabezas-Rodríguez, Unai Martín

**Affiliations:** 1Department of Sociology and Social Work, University of the Basque Country (UPV/EHU), 48940 Leioa, Spain; yolanda.gonzalezr@ehu.eus (Y.G.-R.); unai.martin@ehu.eus (U.M.); 2Research Group on Social Determinants of Health and Demographic Change—OPIK, 48940 Leioa, Spain

**Keywords:** childhood, social inequalities, health-related behaviors, housing conditions, confinement, COVID-19, equity

## Abstract

The COVID-19 lockdown was imposed in a context of notable inequalities in the distribution of the social determinants of health. It is possible that the housing conditions in which children and their families experienced the confinement, and the adoption of healthy behaviors, may have followed unequal patterns. The aim was to describe social inequalities in housing conditions and in health-related behaviors among children during the lockdown in Spain. This cross-sectional study was based on data from an online survey collecting information on the child population (3–12 years) living in Spain (*n* = 10,765). The outcome variables used were several housing conditions and health-related behaviors. The socioeconomic variables used were financial difficulties and parents’ educational level. Crude prevalence and prevalence ratios estimated using Poisson models were calculated. During lockdown, children from families with low educational levels and financial difficulties not only tended to live in poor housing conditions, but were also exposed to negative health determinants such as noise and tobacco smoke; they took less physical exercise, had a poorer diet, spent more time in front of screens and had less social contact. A notable social gradient was found in most of the variables analyzed. The results point to the need to incorporate the perspective of equity in the adoption of policies in order to avoid the increase of pre-existing social inequalities in the context of a pandemic.

## 1. Introduction

The global COVID-19 pandemic led to the introduction of extraordinary measures in Spain, which severely restricted mobility and business, cultural and recreational activities, and closed down schools, universities, hotels, bars and restaurants [1]. On 14 March 2020, the government declared a state of emergency and the period of generalized confinement began. Whereas adults were allowed to leave their homes to carry out certain essential activities such as shopping, in the case of the population under 14 years of age the confinement was total. This situation lasted for six weeks until 25 April, when new legislation was passed to allow children to go out at certain times of day, albeit still under significant restrictions [2].

Several studies have reflected the impact that the measures for managing the epidemic, including lockdown, have had on the health of the child population. They have reported reductions in physical activity, increases in screen time, and deteriorations in diet and sleep quality [3,4,5,6,7]—all factors that may have negative effects on health such as an increase in obesity and impairment of cardiorespiratory capacity [8,9]. In addition, given the importance of social ties to psychological wellbeing and healthy behavior [10,11], a lockdown period that limits all social relationships to the home and the immediate family and disrupts all contact with friends and the school environment may have an impact on children’s mental health as well.

Taking the model of the determinants of social inequalities in health proposed by the Spanish government as our starting point [12], it seems natural to think that the impact of a measure such as confinement on children’s health will vary according to socioeconomic status and, as a result, COVID-19 controls are likely to increase social inequalities in childhood health. The lockdown occurred in a context of pre-existing socioeconomic and health inequalities, in which the inequalities generated by the epidemic exacerbate pre-existing socioeconomic and health inequalities, in what some authors have termed syndemic pandemic [13]. In this sense, the greatest impact of lockdown would have been felt among the most socially disadvantaged population, given that housing conditions affect many aspects of children’s development [14], and that substandard housing can negatively impact their physical, psychological and social well-being [15]. In this regard, the quality and characteristics of the home are key drivers of inequalities [16], which in Spain presents a marked socioeconomic gradient [17]. According to the national Living Conditions Survey, children in poverty are four times more likely to experience extreme heat or cold in their homes, and are more likely to live in buildings with leaking roofs and pipes, with excessive noise and without adequate light [18]. Thus, confinement for those who live in poor quality homes implies greater exposure to dampness and mold, which in turn increase rates of respiratory diseases and allergies [15], as well as noise, scarce natural light and overcrowding, which are all associated with greater risks of depression, anxiety and isolation [15,19]. Living in limited space or in overcrowded conditions has a direct impact on children’s physical and mental health [15], and may also have an indirect impact on other aspects of their lives as well, for example, their performance at school [20].

As regards health-related behaviors, confinement has also entailed changes that have deepened the pre-existing socioeconomic inequalities [21]. It is known that children from more disadvantaged families take less physical exercise [22], have a less healthy diet [23], higher screen time [24], and greater exposure to tobacco smoke [25]. There are also differences according to sex, since the prevalence of physical exercise is higher in boys than in girls, and varies more depending on socioeconomic level among girls; screen time is higher in boys, and is also more dependent on socioeconomic level in boys than in girls [26]. In fact, some studies have already shown the unequal impact of measures such as confinement on physical activity and screen time in the child and adolescent population [27,28,29,30]. However, few studies have addressed inequalities in the living conditions of the child population during lockdown and their impact on social inequalities in health, and those that are available have tended to focus on specific determinants or populations.

The aim of the present article is to describe the social inequalities in the social determinants of children’s health during the COVID-19 lockdown in Spain, applying a comprehensive perspective that explores both the housing conditions in which they live and their health-related behaviors.

## 2. Materials and Methods

### 2.1. Study Design and Participants

Cross-sectional study of the population between 3 and 12 years old living in Spain during the period of confinement was carried out based on data from an online survey administered to families with minors in this age group. The restriction to that age group was due to the use of same valid indicators for the study of determinants of health in the entire age range, as well as the intention to exclude children at the secondary educational level. For the analysis, children self-classified as non-binary were excluded due to small numbers (*n* = 50), as well as children without information regarding socioeconomic variables.

The data were obtained using a questionnaire designed and created in stages. In the first stage, based on the model of the Spanish Commission to Reduce Social Inequalities in Health [12], we carried out a review of the literature in order to identify the social determinants of health that might be affected by confinement. The determinants identified were validated through a survey directed at social and health practitioners and specialists who work with the child population (*n* = 310), recording their assessments of the importance of each determinant and any proposals they may have had for incorporating new ones. The characteristics of this study and its results have been published elsewhere [31]. Based on the results, the final version of the questionnaire was prepared, which also contained socioeconomic variables and family characteristics. The questionnaire was presented in the four official languages in Spain.

### 2.2. Sampling and Data Collection

A non-probability sampling method was applied with self-selection of the participants. Various recruitment strategies were used, including the snowball technique in which families with children of the ages under study were contacted and invited to participate in the study, and were also asked to forward the information to other families. The main means of dissemination of the survey was the social network WhatsApp. Information regarding the study was sent by email to various organizations asking them to publicize it. In order to avoid the possible socioeconomic bias inherent in the use of online administration, a social organization participating in the study contacted families at risk of social exclusion to encourage them to take part, and to help them complete the questionnaire by telephone if necessary. The social network Twitter was also used, creating several tweets that obtained 5211 impressions, 30 retweets and 53 links to the survey. Cases with incomplete information were excluded from the final sample. A total sample of 10,765 children was obtained (*n* boys = 5554; *n* girls = 5211), with a 99% of confidence level and a margin of error of 1.25%. The field work was carried out between 4 and 15 April 2020 through an online questionnaire using the encuestafacil.com platform.

### 2.3. Variables

Housing conditions and health-related behaviors were the health determinant variables used in this study. The housing conditions considered were the lack of outdoor spaces such as balconies, terraces, patios or gardens, the presence of humidity, the scarcity of natural light, high noise levels, and exposure to tobacco smoke in the home (all variables were yes/no questions). Health-related behaviors comprised physical activity, fruit and vegetable intake, excessive consumption of processed or ultra-processed foods (all split into rarely or never vs. two or three times a week, hardly every day or at least once a day), screen time of six hours a day or more, and the scarcity of contact with relatives not living in their household and/or friends (rarely, never or two or three times a week vs. hardly every day or at least once a day). The socioeconomic variables used were household financial difficulties (very difficult to make ends meet, relatively difficult, relatively easy or very easy) and the parents’ highest level of education (primary, secondary or university studies).

### 2.4. Statistical Analysis

Crude prevalence was calculated for the sample description in relation to the socioeconomic variables of the children and their households, as well as the variables related to housing conditions and health-related behaviors. To analyze the bivariate association between health determinants and socioeconomic status variables, prevalence ratios were estimated using Poisson models with robust variance and their 95% confidence intervals (95% CI). Prevalence ratios were calculated, instead of odds ratios due to their easier interpretation as a probability of being at risk between two groups. All the analyses were weighted using the data for level of studies from an official survey representative of the entire Spanish population in order to mitigate the effect of the lower participation of people with lower levels of education. All analyses were carried out separately for boys and girls using SPSS v.26 (IBM).

## 3. Results

The sociodemographic characteristics and the prevalences of each health determinant are shown in Table 1. Mean ages were 6.7 years for boys and 6.6 years for girls. Slightly more than 40% of the sample were aged between 3 and 5 years, around 31% between 6 and 8 around 27–28% between 9 and 12. In all, 3.1% of boys and 2.2% of girls lived in households with severe financial difficulties, slightly more than one in four in households with some financial difficulties, more than 60% in relatively affluent families and 8–9% in wealthy families. More than 51% of parents had completed secondary school, around 44.5% were university graduates and between 3.7% and 4.5% had only completed primary school.

Regarding the housing conditions in which the children lived during lockdown, 26% had no access to outdoor spaces such as balconies, terraces, patios or gardens, around 12% lived in damp conditions, and around 9% had scarce natural light. More than 4% lived in excessively noisy environments and more than 30% were exposed to tobacco smoke. As regards health-related behaviors, 18.4% of the girls and 21.8% of the boys rarely or never took physical activity, 17.6% of the girls and 20.1% of the boys had insufficient fruit intake, a figure that rose above 41% in both sexes in the case of vegetables, and 31.4% of boys and 29% of girls presented excessive consumption of processed or ultra-processed foods. Around 25% of the children spent more than six hours a day in front of screens, and 44.5% of boys and 38.7% of girls had infrequent contact with relatives not living in their household and/or friends.

Table 2 shows the prevalence of health determinants according to the household’s financial situation and parents’ level of education. Notable inequalities according to parents’ level of education emerged in all the variables analyzed in both boys and girls, with the exception of the consumption of processed or ultra-processed foods and the lack of outdoor space. The social gradient is clear; inequalities were recorded along the entire social scale, not only between the extremes. The prevalence of determinants such as lack of natural light, presence of dampness, exposure to tobacco smoke in the home, lack of physical activity or insufficient consumption of fruit was more than twice as high in disadvantaged families than in their socioeconomically favored peers. For instance, 48.2% of boys who live in homes undergoing financial hardship were exposed to tobacco smoke, compared to 21.8% of those from families without financial difficulties, and 41.7% of girls whose parents had primary studies were exposed to tobacco smoke compared with 22.3% of those born to university graduates. Screen exposure also presented major differences of almost 19 percentage points for boys and 15 points for girls depending on parents’ financial situation, and of almost 17 points for boys and 24 points for girls in relation to parents’ level of education.

Table 3 shows the prevalence ratios of the determinants of children’s health according to their parents’ financial situation. With regard to housing conditions, households with greater financial difficulties were more likely to live in homes with poor structural characteristics, two or even three times more likely to suffer excessive noise or damp, 50% more likely to have no access to outdoor spaces, and 94% more likely to be exposed to tobacco smoke. As for health-related behaviors, in boys there was a clear social gradient to the detriment of households with fewer resources in terms of insufficient physical activity and fruit intake, and the probability of prolonged screen exposure was 44% higher in homes with severe financial hardship. These patterns of physical activity and fruit and vegetable intake were reproduced among girls. Lastly, boys and girls from households with difficulties or severe financial difficulties were around 40% more likely to have low levels of contact with family and friends than their financially better-off peers.

Turning now to parents’ level of education (Table 4), a clear social gradient was observed in both boys and girls in practically all the variables studied. The most pronounced inequalities between educational extremes were observed in the presence of high levels of noise in the home, which was greater than four times more frequent in boys and three times more frequent in girls with less educated parents, and in the presence of damp, twice as likely in boys and 50% more likely in girls; this inequality was also observed between children of parents who completed secondary school and those whose parents were university graduates. Similarly, there was a greater risk of exposure to tobacco smoke in the home in less educated families. In the case of health-related behaviors, the children of less educated parents were between two and three times more likely to consume insufficient amounts of fruit and vegetables, and were more likely to spend more than six hours in front of screens and to have little contact with family and/or friends. No statistically significant differences were found in the availability of outdoor space and consumption of processed or ultra-processed foods between the most and the least educated families.

## 4. Discussion

The social determinants of children’s health presented an unequal distribution during confinement. Boys and girls from more disadvantaged families lived in homes with worse structural characteristics and were more exposed to health determinants such as noise or tobacco, performed less physical activity, had a poorer diet, higher screen time and less social contact.

Several recent studies have highlighted the impact of pandemic management measures, and especially lockdown, on determinants of children’s health such as physical activity, diet and sleep [3,4,5,7]. Although few of these studies have specifically analyzed whether this impact is affected by socioeconomic status, their results broadly coincide with ours. A study carried out in Navarre in Spain [30] demonstrated the impact of confinement in reducing physical activity in the population aged 8 to 16 years, and stressed that this reduction was greater if the child’s mother was of foreign origin and had not studied at university. However, perhaps due to its small sample size, the Navarre study did not find significant differences in certain habits in which our study observed an impact of socioeconomic status, such as screen time and diet. Although without specifically addressing lockdown, another study carried out in Canada showed changes in physical activity, sedentary habits and sleep during the pandemic: the study found that, in lower-income households, a greater proportion of children and adolescents reduced their physical activity and increased their screen time and their use of social networks, while those from high-income households were more likely to increase their physical activity both inside and outside the home (e.g., walking or biking) [27]. Using data from the same study, Moore et al. [28] found that parents’ low educational attainment was associated with a decrease in exercise outside the home and an increase in sedentary habits. Similarly, a study in Bosnia–Herzegovina indicated that adolescents with more educated parents were more likely to take physical exercise during the first phase of the COVID-19 pandemic (March–April) than those whose parents had a lower level of studies [29].

To our knowledge, no previous studies have assessed housing conditions in relation to the 2020 lockdown. Pre-COVID era research established the relationship between time spent at home and health, highlighting the greater risk of exposure to tobacco smoke with increasing time at home [32] and its negative effect on children’s respiratory health [33]. Likewise, prolonged exposure to dampness and mold increases the risk of allergic and respiratory diseases [17], especially asthma [19]. Noisy environments and the scarcity of natural light have been associated with higher levels of anxiety and depression [15], and the insufficient provision of space may affect children’s physical and mental health [15,19]. Our results indicate that many children are exposed to situations of this kind, especially those in the most vulnerable social groups. On top of the pre-COVID era evidence showing that minors from socially less privileged families have a poorer state of health in general [21,34], the confinement in poor-quality housing could have led to a deeper accumulation of risks that might further affect their health.

The existence of social inequalities in health and in health determinants is due to structural factors that condition the unequal exposure to intermediate factors related to health. This exposure depends to a large extent on social status, which is constructed in turn on the basis of variables such as gender, level of education and socioeconomic status. The unequal impact of lockdown on health in distinct sectors of society is also due to the structural characteristics of our economic and political system and its macroeconomic, labor and welfare policies. This implies that families with lower educational attainment have more limited access to housing, poorer working conditions and lower income, which, together with other factors, have caused the unequal impact of confinement on the social determinants of children’s health described in our study.

The study has certain limitations that should be borne in mind in the interpretation of the results. The main limitation results from the type of sampling used. Even when based on large samples, the self-selection technique may yield biased results compared to probability sampling due to factors such as the underrepresentation of less favored populations or less extreme opinions [35]. However, the study sought to address the problem of the lower participation of the most vulnerable families by providing telephone support for this population in the data collection and by using a specific weighting system. Another possible source of bias is the use of key informants, i.e., the father, mother or legal guardian of the population under 13 years of age, rather than with the child him/herself, but the questions were designed accordingly. Among the study’s strengths are the sample size (*n* = 10,765), larger than most studies of this type, and the timing of the field work and the circumstances in which it was carried out; in a complex research context, we were able to measure an entirely new phenomenon which has had a major impact on social and health inequalities in a highly vulnerable population.

## 5. Conclusions

Our study has several important political and public health implications. First, it highlights the potential impact of confinement on the health and health inequalities in the child population. Knowing that health inequalities in this population have major repercussions for their present and future health, the results of the study allow us to consider the impact of the COVID-19 pandemic and its management on children’s health, beyond infection and mortality rates; indeed, they add an important extra facet to be included in the assessment of the management of the epidemic. Second, the impact of lockdown on health and health inequalities underlines the need to incorporate the concept of health equity, both in the management of the COVID-19 epidemic and beyond. Taking account of issues of equity is essential in order to improve the design of prevention measures to face similar situations in the future without compromising other aspects of public health.

In this sense, the measures to control the pandemic must be accompanied by measures that allow and promote physical activity for children and the maintenance of spaces for sociability among peers, such as the conditioning of parks and public spaces aimed at this group. Likewise, the maintenance of face-to-face school activity should be prioritized, which favors healthy food intake, as well as physical exercise and social relationships, and reduces the time spent on screens. In addition, from a more structural approach, improving the housing conditions of families, especially with young children, is an essential public health measure, especially in times of restricted mobility outdoors. Our results, which show the need to incorporate equity in the adoption of universal policies so that they do not generate greater inequality, goes beyond the framework of COVID-19 management and shows the importance of this perspective in setting public health policies.

In conclusion, our study shows the importance of taking into account not only the health impacts of the confinement beyond those that occur directly due to the virus, but also the need to apply an equity approach to the issue of children’s health. Equity is a key element for improving the health of the child population and for making political decisions that minimize the effects of prevention measures against COVID-19 or future pandemic situations.

## Figures and Tables

**Table 1 ijerph-18-04087-t001:** Sociodemographic characteristics of the sample and prevalence of the health determinants analyzed according to sex.

Variables Title	Boys (*n* = 5554)	Girls (*n* = 5211)	*p*-Value *^a^*
Mean age (SD)	6.7 (2.8)	6.6 (2.8)	
*Age groups*			0.341
3–5 years	40.5%	41.8%	
6–8 years	31.0%	30.8%	
9–12 years	28.4%	27.4%	
*Household’s financial situation*			<0.001
Very difficult to make ends meet	3.1%	2.2%	
Relatively difficult	25.2%	27.8%	
Relatively easy	63.6%	61.2%	
Very easy	8.1%	8.8%	
*Parents’ level of education*			0.084
Primary studies	4.5%	3.7%	
Secondary studies	51.0%	51.9%	
University studies	44.5%	44.4%	
Lack of outdoor space	26.0%	26.2%	0.771
Presence of dampness	12.2%	11.6%	0.323
Lack of natural light	8.9%	8.7%	0.660
Noise	4.0%	4.5%	0.222
Exposure to tobacco smoke	31.3%	30.6%	0.402
Lack of physical activity	21.8%	18.4%	<0.001
Insufficient fruit intake	20.1%	17.6%	<0.001
Insufficient vegetable intake	41.6%	41.9%	0.762
Excessive consumption of processed or ultraprocessed foods	31.4%	29.0%	0.009
Screen exposure ≥ 6 h	25.3%	24.1%	0.136
Low level of contact with family and friends	44.5%	38.7%	<0.001

*^a^ p*-values for chi-square statistics.

**Table 2 ijerph-18-04087-t002:** Prevalence of housing conditions and health-related behaviors according to the household’s financial situation and parents’ educational level.

Type of Health Determinant	Variables	Very Difficult to Make Ends Meet	Relatively Difficult	Relatively Easy	Very Easy	*p*-Value *^a^*	Primary Studies	Secondary Studies	University Studies	*p*-Value *^a^*
**Boys**										
*Housing conditions*	Lack of outdoor spaces	32.4%	31.6%	24.0%	21.3%	<0.001	29.1%	27.0%	24.5%	0.053
	Presence of dampness	22.5%	18.1%	9.9%	7.8%	<0.001	22.7%	12.9%	10.5%	<0.001
	Lack of natural light	22.4%	12.0%	7.7%	4.2%	<0.001	13.9%	9.6%	7.6%	<0.001
	Noise	11.8%	8.2%	2.0%	2.9%	<0.001	11.6%	4.6%	2.6%	<0.001
	Exposure to tobacco smoke	48.2%	43.3%	26.9%	21.8%	<0.001	51.4%	37.5%	22.2%	<0.001
*Health-related behaviors*	Lack of physical activity	35.3%	29.0%	19.5%	13.3%	<0.001	35.1%	24.1%	17.9%	<0.001
	Insufficient fruit intake	25.4%	27.5%	18.0%	10.9%	<0.001	39.8%	24.2%	13.3%	<0.001
	Insufficient vegetable intake	44.7%	48.6%	40.0%	31.9%	<0.001	65.7%	47.8%	32.1%	<0.001
	Excessive consumption of processed or ultraprocessed foods	38.2%	32.2%	30.8%	29.9%	0.160	27.5%	32.5%	30.4%	0.095
	Screen exposure ≥ 6 h	41.2%	29.4%	23.3%	22.5%	<0.001	35.3%	30.3%	18.6%	<0.001
	Low level of contact with family and friends	55.3%	51.4%	41.8%	38.6%	<0.001	46.2%	46.7%	41.7%	0.001
**Girls**										
*Housing conditions*	Lack of outdoor spaces	24.8%	31.4%	24.5%	23.5%	<0.001	22.4%	27.7%	24.9%	0.038
	Presence of dampness	18.4%	19.0%	9.0%	5.0%	<0.001	15.1%	13.6%	9.0%	<0.001
	Lack of natural light	20.4%	12.6%	7.5%	1.8%	<0.001	10.4%	9.6%	7.4%	0.016
	Noise	11.5%	8.3%	2.9%	1.8%	<0.001	8.3%	5.6%	2.9%	<0.001
	Exposure to tobacco smoke	35.1%	42.7%	26.2%	20.0%	<0.001	41.7%	36.9%	22.3%	<0.001
*Health-related behaviors*	Lack of physical activity	27.4%	22.0%	17.4%	11.6%	<0.001	20.5%	20.5%	15.7%	<0.001
	Insufficient fruit intake	32.7%	23.4%	15.5%	11.0%	<0.001	37.2%	20.5%	12.7%	<0.001
	Insufficient vegetable intake	54.5%	49.0%	40.1%	29.6%	<0.001	62.8%	50.4%	30.2%	<0.001
	Excessive consumption of processed or ultraprocessed foods	40.4%	32.2%	27.5%	26.3%	<0.001	35.3%	29.1%	28.4%	0.136
	Screen exposure ≥ 6 h	34.8%	28.8%	22.0%	20.3%	<0.001	41.9%	28.2%	17.8%	<0.001
	Low level of contact with family and friends	48.7%	46.8%	35.6%	33.3%	<0.001	54.7%	41.1%	34.7%	<0.001

*^a^**p*-values for chi-square statistics.

**Table 3 ijerph-18-04087-t003:** Prevalence ratios and 95% CIs for housing conditions and health-related behaviors according to the household’s financial situation.

Type of Health Determinant	Variables	Boys			Girls		
		Very Difficult to Make Ends Meet	Relatively Difficult	Easy	Very Difficult	Relatively Difficult	Easy
*Housing conditions*	Lack of outdoor spaces	1.53 (1.11–2.10) *	1.51 (1.19–1.93) *	1.13 (0.89–1.43)	0.89 (0.58–1.36)	1.29 (1.03–1.61) *	1.00 (0.81–1.25)
	Presence of dampness	2.36 (1.48–3.76) *	1.87 (1.28–2.75) *	1.03 (0.70–1.51)	3.00 (1.66–5.44) *	2.93 (1.85–4.64) *	1.44 (0.91–2.29)
	Lack of natural light	6.30 (3.22–12.33) *	3.29 (1.76–6.15) *	2.13 (1.14–3.96) *	10.27 (3.94–26.78) *	7.12 (2.95–17.15) *	4.22 (1.75–10.17) *
	Noise	3.10 (1.52–6.23) *	2.11 (1.15–3.86) *	0.46 (0.24–0.87) *	3.86 (1.51–9.87) *	3.30 (1.56–7.00) *	1.16 (0.54–2.49)
	Exposure to tobacco smoke	1.94 (1.51–2.48) *	1.68 (1.37–2.06) *	1.13 (0.92–1.39)	1.37 (0.99–1.89)	1.71 (1.39–2.10) *	1.11 (0.91–1.36)
*Health-related behaviors*	Lack of physical activity	2.87 (1.98–4.16) *	2.35 (1.71–3.24) *	1.62 (1.18–2.23)	2.52 (1.57–4.03) *	2.09 (1.46–2.99) *	1.73 (1.21–2.46) *
	Insufficient fruit intake	1.94 (1.30–2.91) *	2.19 (1.60–3.01) *	1.57 (1.15–2.14)	2.78 (1.84–4.21) *	1.98 (1.43–2.75) *	1.37 (0.99–1.89)
	Insufficient vegetable intake	1.09 (0.88–1.36)	1.21 (1.04–1.41) *	1.06 (0.91–1.22)	1.63 (1.28–2.07) *	1.50 (1.26–1.78) *	1.27 (1.07–1.50) *
	Excessive consumption of processed or ultraprocessed foods	1.21 (0.94–1.56)	0.99 (0.82–1.19)	0.96 (0.80–1.15)	1.72 (1.26–2.35) *	1.33 (1.06–1.66) *	1.16 (0.93–1.45)
	Screen exposure ≥ 6 h	1.44 (1.11–1.88) *	1.08 (0.88–1.32)	0.88 (0.72–1.08)	1.52 (1.08–2.14) *	1.28 (1.02–1.62) *	1.05 (0.83–1.31)
	Low level of contact with family and friends	1.41 (1.15–1.74) *	1.35 (1.15–1.58) *	1.09 (0.93–1.27)	1.43 (1.09–1.87) *	1.46 (1.22–1.74) *	1.09 (0.91–1.31)

Reference: Very easy to make ends meet. “*” *p* < 0.05.

**Table 4 ijerph-18-04087-t004:** Prevalence ratios and 95% CIs for housing conditions and health-related behaviors according to the parents’ educational level.

Type of Health Determinant	Variables	Boys		Girls	
		Primary Studies	Secondary Studies	Primary Studies	Secondary Studies
*Housing conditions*	Lack of outdoor spaces	1.21 (0.98–1.50)	1.13 (1.01–1.26) *	0.90 (0.68–1.19)	1.11 (0.99–1.24)
	Presence of dampness	2.28 (1.73–3.00) *	1.28 (1.07–1.54) *	1.51 (1.04–2.17) *	1.34 (1.11–1.61) *
	Lack of natural light	1.88 (1.32–2.68) *	1.28 (1.03–1.58) *	1.25 (0.79–1.97)	1.18 (0.96–1.47)
	Noise	4.61 (2.89–7.35) *	1.77 (1.23–2.55) *	3.21 (1.80–2.21) *	2.21 (1.53–3.20) *
	Exposure to tobacco smoke	1.98 (1.71–2.30) *	1.44 (1.30–1.59) *	1.42 (1.18–1.71) *	1.25 (1.13–1.38) *
*Health-related behaviors*	Lack of physical activity	1.81 (1.48–2.20) *	1.23 (1.09–1.39) *	1.27 (0.94–1.72)	1.26 (1.09–1.45) *
	Insufficient fruit intake	2.47 (2.04–2.99) *	1.49 (1.30–1.70) *	2.54 (2.02–3.18) *	1.40 (1.20–1.63) *
	Insufficient vegetable intake	1.78 (1.59–1.99) *	1.21 (1.20–1.40) *	1.92 (1.68–2.19) *	1.54 (1.41–1.68) *
	Excessive consumption of processed or ultraprocessed foods	0.87 (0.70–1.07)	1.03 (0.94–1.13)	1.19 (0.96–1.46)	0.99 (0.89–1.09)
	Screen exposure ≥ 6 h	1.80 (1.48–2.19) *	1.54 (1.36–1.73) *	2.22 (1.81–2.71) *	1.49 (1.31–1.69) *
	Low level of contact with family and friends	1.12 (0.97–1.29)	1.13 (1.05–1.22) *	1.63 (1.41–1.90) *	1.23 (1.12–1.34) *

Reference: University studies. “*” *p* < 0.05.

## Data Availability

Data used during the current study are available from the corresponding author on reasonable request.

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
