# Peer review of "Social Inequalities in Health Determinants in Spanish Children during the COVID-19 Lockdown"

_ijerph, 2021, doi:10.3390/ijerph18084087_

Round 1

Reviewer 1 Report

It is a very interesting research with informative findings, and equally well-presented. Just a little suggestion

For the benefit of your readers that may not be well-informed in statistical analysis, I suggest you explain further the statistical analysis part, ie line 129-137. Somethings like 'what crude prevalence mean, why Poisson model was used as against other statistical models, and what 95% CI mean'. 

I am sure this will give a better apprehension of the paper to all readers and not only readers with statistical backgrounds.

Author Response

Thank you for your revision of the manuscript and for considering interesting.

We agree that some statistical vocabulary may not be of common knowledge of all potential readers. We have clarified the use of prevalence ratios with Poisson model, in contrast with other type of analysis to measure association between one dichotomous dependent variable and the independent variables.

Reviewer 2 Report

Thank you for the opportunity to review this article. The work is interesting, but some aspects should be taken into account before publication.

Comments and suggestions for Authors:

This paper confirms the impact of confinement on the health and health inequalities in the child population. Since this is a public healthy journal, I want to see more of a focus on the discussion about potential public health measures that can be done in Spain to help the issue.

Also the paper needs to explain why only population aged 3-12 were included and why not older ages. There is data on older children about this issue. The researchers should include this in their paper.

The paper did not mention the inclusion and exclusion criteria. Please discuss power calculation and how the sample size is adequate. This needs to be stated clearly as well. 

Author Response

Thank you for your valuable revision of the manuscript. Below you can find our point-by-point response (our responses in cursive) to the comments.

Response point-by-point:

Point 1: This paper confirms the impact of confinement on the health and health inequalities in the child population. Since this is a public healthy journal, I want to see more of a focus on the discussion about potential public health measures that can be done in Spain to help the issue.

We agree that a more specific explanation about potential health measures that can be implemented in Spain can increase the contribution of the article. We have include it in lines 319-326 of the new version of the manuscript.

Point 2: Also the paper needs to explain why only population aged 3-12 were included and why not older ages. There is data on older children about this issue. The researchers should include this in their paper.

A deeper explanation about the selected age group population has been included in lines 90-92.

Point 3: The paper did not mention the inclusion and exclusion criteria.

The exclusion criteria for the analysis has been clarify in lines 93-95 and 120.

Regarding the criteria for including or excluding children in the sample, it is explained in section 2.1 and 2.2 of the manuscript. As the sampling was done with self-selection of the participants, the definition of the population under study done is considered the criteria.

Point 4: Please discuss power calculation and how the sample size is adequate. This needs to be stated clearly as well. 

A specification about the confidence level and margin of error of our sample size has been included in lines 121-122.

Reviewer 3 Report

In abstract, the “determinants of health” sentence (line 15) is confusing because the socioeconomic variables are also determinants of health. Do the authors mean the dependent/outcome variables?

The introduction is well-written and provides a compelling case rationale for the research study. In line 82-83, the authors that that there are looking at social inequalities in the social determinants of health—is this referring to housing conditions and health behaviors? If so, health behaviors are not broader social determinants—they are individual-level determinants. Perhaps the authors could remove the word “social” and just use the term “determinants of health”?

The first part of the Materials and Methods section (lines 88-90) is not a complete sentence.

In line 91, how is the survey different from a questionnaire? This sentence is confusing as written.

In line 93, do the authors mean “review of the literature”, as opposed to “review of the bibliography"?

Despite the biases associated with using a non-probability sample, it is admirable that the researchers worked with a social organization to reach out to families who may not have internet access.

In line 120, the authors need to elaborate on what “the presence of damp” means. Damp what?

Overall, the authors need to expand on how the various measures were operationalized. How were the housing condition variables operationalized—yes versus no? How was tobacco smoke at home operationalized—was this tobacco versus no tobacco, or where there different levels of tobacco smoke that were explored? The authors need to specify how each variable was operationalized. The authors need to also specify the reference categories for each health behavior variable (e.g. “very often” for physical activity?). Finally, the authors need to describe how the socioeconomic variables were each operationalized.

The statistical analysis section needs to be expanded as well. Did the authors conduct only bivariate analysis, or did they also conduct multivariate analyses? If multivariate analyses were conducted, the authors need to describe the main models they examined. If the authors only used bivariate analyses, it is concerning because the researchers likely did not account for important confounding factors.

Housing conditions and parental education are likely highly co-linear; therefore, by only conducting bivariate analyses, the authors have not truly teased out the independent effects of each of these variables.

In line 214, the authors use the term “social determinants of health” again – this needs to be changed to just determinants of health throughout the article. Health behaviors are not social determinants of health—actually, it is confusing because education IS a social determinant of health.

A significant limitation of the study is its lack of multivariate analysis. The authors either need to conduct this analysis and redo the Methods and Results sections or justify why they did not conduct such an analysis, especially when they have the benefit of such a large sample size.

Author Response

Thank you for your valuable revision of the manuscript. Your thoughtful comments certainly help to improve the manuscript. Below you can find our point-by-point response (our responses in cursive) to the comments.

Response point-by-point:

Point 1: In abstract, the “determinants of health” sentence (line 15) is confusing because the socioeconomic variables are also determinants of health. Do the authors mean the dependent/outcome variables?

We agree with your comment, it could be confusing. We have put “outcome variables” instead of “determinants of health”.

Point 2: The introduction is well-written and provides a compelling case rationale for the research study. In line 82-83, the authors that that there are looking at social inequalities in the social determinants of health—is this referring to housing conditions and health behaviors? If so, health behaviors are not broader social determinants—they are individual-level determinants. Perhaps the authors could remove the word “social” and just use the term “determinants of health”?

The definition of the social determinants of health used in the article is based on the conceptual framework developed by the Commission on Social Determinants of Health  (CSDH)  of the World Health Organization (WHO) in 2010. In that framework, the behavioral factors are included as social determinants of health. The link to the referred document is: https://apps.who.int/iris/bitstream/handle/10665/44489/9789241500852_eng.pdf?sequence=1&isAllowed=y

According to this framework of the CSDH of the WHO, we have use the word social when including both types of social determinants of health, the housing conditions and the health-related behaviors, and the specific type when referring to each one.

Point 3:The first part of the Materials and Methods section (lines 88-90) is not a complete sentence.

We complete the sentence adding “was carried out”.

Point 4: In line 91, how is the survey different from a questionnaire? This sentence is confusing as written.

Although “survey” and “questionnaire” do not mean the same, being the survey the research technique used and the questionnaire the instrument for collecting data, we have deleted the word “survey” as it could be unnecessary in this sentence.

Point 5: In line 93, do the authors mean “review of the literature”, as opposed to “review of the bibliography"?

We agree with the change and we have put “review of the literature” instead of “review of the bibliography".

Point 6: Despite the biases associated with using a non-probability sample, it is admirable that the researchers worked with a social organization to reach out to families who may not have internet access.

Thank you for your comment. We were aware of that issue as a potential bias of the sampling, so a great effort was done to try to minimalize.

Point 7: In line 120, the authors need to elaborate on what “the presence of damp” means. Damp what?

Damp or dampness as a synonym of humidity. We have changed for dampness throughout the article.

Point 8: Overall, the authors need to expand on how the various measures were operationalized. How were the housing condition variables operationalized—yes versus no? How was tobacco smoke at home operationalized—was this tobacco versus no tobacco, or where there different levels of tobacco smoke that were explored? The authors need to specify how each variable was operationalized. The authors need to also specify the reference categories for each health behavior variable (e.g. “very often” for physical activity?). Finally, the authors need to describe how the socioeconomic variables were each operationalized.

We agree with your comment. In section 2.3 Variables, for housing conditions variables we have added, “all variables were yes/no questions”. For health behaviors, we have clarified the splitting of the categories. For socioeconomic variables, we used the original categories, without adding them or splitting into groups, but we have added the explanation in methods section.

Point 9: The statistical analysis section needs to be expanded as well. Did the authors conduct only bivariate analysis, or did they also conduct multivariate analyses? If multivariate analyses were conducted, the authors need to describe the main models they examined. If the authors only used bivariate analyses, it is concerning because the researchers likely did not account for important confounding factors. Housing conditions and parental education are likely highly co-linear; therefore, by only conducting bivariate analyses, the authors have not truly teased out the independent effects of each of these variables.

According to our research objectives, we only did bivariate analysis, between each outcome variable and the two socioeconomic variables separately. We have included the word “bivariate” to avoid misunderstandings. The analysis strategy responds to the objectives of the study, that is, to analyze the differences in the determinants according to the independent variables, so we did not adjust or delete the effect of other variables.

Regarding the possible co-linear of variables, if so, does not affect the analysis because we did not want to analyze the independent effects.

Point 10: In line 214, the authors use the term “social determinants of health” again – this needs to be changed to just determinants of health throughout the article. Health behaviors are not social determinants of health—actually, it is confusing because education IS a social determinant of health.

Please, see the response to the second comment of this revision.

Point 11: A significant limitation of the study is its lack of multivariate analysis. The authors either need to conduct this analysis and redo the Methods and Results sections or justify why they did not conduct such an analysis, especially when they have the benefit of such a large sample size.

We did not conduct multivariate analysis because it was out of focus of the research objectives of the study presented in this article. We think that it is not necessary to make a justification of the absence of a type of analysis in the Methods section.

Round 2

Reviewer 3 Report

The authors have satisfactorily addressed my previous comments. Thank you for your important contribution!